# Vigilant Attention, Cerebral Blood Flow and Grey Matter Volume Change after 36 h of Acute Sleep Deprivation in Healthy Male Adults: A Pilot Study

**DOI:** 10.3390/brainsci12111534

**Published:** 2022-11-12

**Authors:** Han-Rui Zeng, Fan Xu, Jing Zhang, Qiong-Fang Cao, Yu-Han Wang, Peng Zhang, Yong-Cong Shao, Shao-Ping Wu, Xie-Chuan Weng

**Affiliations:** 1Department of Clinic Medicine, Chengdu Medical College, Chengdu 610500, China; 2Department of Public Health, Chengdu Medical College, Chengdu 610500, China; 3MOEMIL Laboratory, School of Optoelectronic Information, University of Electronic Science and Technology of China, Chengdu 610054, China; 4School of Psychology, Beijing Sport University, Beijing 100084, China; 5Department of Neuroscience Beijing Institute of Basic Medical Sciences, Beijing 100850, China

**Keywords:** sleep deprivation, cerebral blood flow, arterial spin labeling, grey matter volume, psychomotor vigilance task

## Abstract

It is commonly believed that alertness and attention decrease after sleep deprivation (SD). However, there are not enough studies on the changes in psychomotor vigilance testing (PVT) during SD and the corresponding changes in brain function and brain structure after SD. Therefore, we recruited 30 healthy adult men to perform a 36 h acute SD experiment, including the measurement of five indicators of PVT every 2 h, and analysis of cerebral blood flow (CBF) and grey matter volume (GMV) changes, before and after SD by magnetic resonance imaging (MRI). The PVT measurement found that the mean reaction time (RT), fastest 10% RT, minor lapses, and false starts all increased progressively within 20 h of SD, except for major lapses. Subsequently, all indexes showed a significant lengthening or increasing trend, and the peak value was in the range of 24 h-32 h and decreased at 36 h, in which the number of major lapses returned to normal. MRI showed that CBF decreased in the left orbital part of the superior frontal gyrus, the left of the rolandic operculum, the left triangular part, and the right opercular part of the inferior frontal gyrus, and CBF increased in the left lingual gyrus and the right superior gyrus after 36 h SD. The left lingual gyrus was negatively correlated with the major lapses, and both the inferior frontal gyrus and the superior frontal gyrus were positively correlated with the false starts. Still, there was no significant change in GMV. Therefore, we believe that 36 h of acute SD causes alterations in brain function and reduces alert attention, whereas short-term acute SD does not cause changes in brain structure.

## 1. Introduction

Sleep deprivation (SD) can cause a variety of health problems and have adverse consequences on life and production. With the rapid development of society and drastic changes in lifestyle, SD has become a serious social problem and one of the hot topics in the field of medical research. Acute SD is more likely to occur in the military, transportation, medical, and other important industries, and acute SD experiments are quite manageable and feasible [1,2], so this kind of research has a high utility value.

Objective examination methods used in SD studies include the psychomotor vigilance test (PVT), medical imaging techniques, electroencephalography examinations, multiple sleep latency tests, and biochemical assessment indicators, with the first two methods being most commonly used in experiments. The PVT was extremely sensitive to slower reaction times, and attention deficits that occur during acute SD, and are valid for measures of sustained attention [3]. Among medical imaging techniques, blood oxygen level-dependent functional magnetic resonance imaging (BOLD-fMRI) can reveal the complex effects of neuronal activity on cerebral blood flow (CBF), blood volume, and oxygen levels [4], indirectly representing neuronal activity. Arterial Spin Labeling-fMRI (ASL-fMRI) can only measure CBF as a physiological indicator. Still, it is simple to operate and more accurate to measure CBF than BOLD-fMRI, and CBF variations are primarily brought on by neuronal activity [5]. However, there are few reports on using ASL in acute SD studies.

In addition to functional brain imaging studies, voxel-based morphometry (VBM) methods have been gradually developed to analyze structural changes in the brain [6]. It can be used to quantitatively measure the volume and composition of gray matter, white matter, and cerebrospinal fluid in vivo, of which gray matter volume (GMV) is the most commonly used measure in VBM analysis [6,7,8]. It is well known that behavioral changes may manifest impaired brain function [9,10], but changes in brain function may be reversible [11]. In contrast, structural brain damage may be challenging to recover from [12]. For this reason, it is crucial to study structural brain changes in subjects who have experienced acute SD. It has been previously reported that acute SD can lead to brain atrophy and a decline in the somatosensory region GMV as the duration of SD increases [8]. Another study showed a significant increase in gray matter density in the right frontal pole, the right superior frontal gyrus, and the right middle frontal gyrus after 24 h of acute SD [13]. Since there are few reports on whether short-term SD can cause minor morphological changes in brain structure, the results need to be verified. We dynamically observed the changes of PVT and CBF in normal adults after 36 h of SD and analyzed the changes of GMV in brain regions by VBM.

## 2. Methods

### 2.1. Participants

A total of 30 healthy men attending medical school were included in the present study. The inclusion criteria were as follows: right-handedness; regular sleeping habits, without sleep disorders; regular eating habits; and without addiction to nicotine, alcohol, or other substances; no intake of any stimulants such as alcohol, caffeine, or drugs 48 h before the experiment; and no prior record of any mental or neurological illness.

This research project was approved by the Institutional Review Board of Beihang University (Beijing, China) prior to initiation, with approval number BM20180040. This experiment was performed under strict compliance with the Declaration of Helsinki. All participants provided written informed consent.

### 2.2. Experiment Paradigm

The experiment was carried out in a comfortable and well-equipped sleep laboratory (Key laboratory for Neuroinformation of Ministry of Education, University of Electronic Science and Technology of China, Chengdu, China). The entire experiment was monitored and accompanied by medical professionals in shifts. After rested wakefulness (RW) (1st day), all subjects underwent a 36 h SD experiment starting at 8: 00 am on the 2nd day and continuing until 8: 00 pm on the 3rd day. Subjects were asked to stay awake for 36 h. Figure 1 shows the experimental process of 36 h acute SD. A PVT test was performed to test their vigilance at different sessions. PVT started from RW at point 1, and repeated every 2 h from point 2 to 19 during SD. Hence, there were 19 checkpoints throughout the experiment. Each subject underwent MRI on two separate occasions: once before normal sleep at 8:00 p.m. on 1st day, and once after acute SD at 8:00 p.m. on 3rd day.

### 2.3. MRI Data Acquisition

The neuroimaging data were acquired using a 3T GE DISCOVERY MR750 scanner (General Electric, Fairfield, CT, USA) equipped with a high-speed gradient. All subjects underwent a resting-state pCASL scan and a high-resolution three-dimensional (3D) T1-weighted sequence scan. The pCASL scanning parameters are shown below: repetition time = 4635 ms; echo time = 10 ms; slice thickness = 4 mm; no gap; flip angle = 111°; voxel size = 1.875 × 1.875 × 4 mm^3^; matrix size = 128 × 128, 36 slices; post-labelling delay = 1.525 s; label duration = 1.45 s; number of excitations = 3; and 269 s. The 3D T1-weighted image was scanned using the following parameters: repetition time = 5.96 ms; echo time = 1.956 ms; slice thickness = 1 mm; no gap; flip angle = 12°; and matrix size = 256 × 256,156 slices. All participants were instructed to lie motionless on the scanning bed, with closed eyes, and to think about nothing in particular. In order to minimize head movement, their head was restrained with a sponge. Earplugs were used to protect against hearing loss. Before the start of each MRI scan, the operator reminded the subjects to stay awake at all times.

### 2.4. Data Processing

SPM8 (http://www.fil.ion.ucl.ac.uk/spm/software/spm8/, accessed on 26 May 2022) package was used to pretreat the ASL data. First, the CBF images were co-registered with the correspondent T1-weighted structural pictures using reciprocal information. Then, co-registered T1 pictures were segmented into grey matter, white matter, and cerebrospinal fluid. Finally, each CBF image was normalized to the space of the Montreal Neurological Institute based on the transformation parameters estimated during the nonlinear co-registration. The resulting CBF images were normalized by deducting the mean CBF image from the grey matter mask and dividing it by the standard deviation of the CBF. The normalized CBF maps were subsequently smoothed with 6 mm full width at half maximum [14].

For 3D-T1 MRI data processing, we used the SPM12 (http://www.fil.ion.ucl.ac.uk/spm/software/spm12/, accessed on 26 May 2022) together with the CAT12 for the SPM (http://www.neuro.uni-jena.de/cat/, accessed on 26 May 2022), for VBM analysis. The VBM analysis included normalization, segmentation, and smoothing. Briefly, the original T1-weighted image of each subject was spatially normalized and partitioned into grey matter, white matter, and cerebrospinal fluid, after which the normalized GMV pictures were smoothed with a Gaussian kernel of 8 mm full width at half maximum.

### 2.5. Cognitive Measurement

PC PVT 2.0 was used for the PVT test, and each PVT test lasted 10 min [15]. The tests were performed on a computer with a 16-inch LCD screen and responded with a highly sensitive mouse. Participants were asked to focus on a red rectangular box with two 1.3° viewing angles in the middle of a black screen. During the test, Arabic numbers appeared at random intervals of 2–10 s in the middle of the screen, and the actual reaction time (RT) was displayed by clicking the left mouse button as soon as it was seen. The test was conducted in a quiet environment, with concentration, and avoidance of other activities during the test. Mean RT, fastest 10% RT, minor lapses, major lapses, and false starts were measured in this experiment.

### 2.6. Sample Size

GPower software version 3.1.97 (http://www.gpower.hhu.de/, accessed on 20 July 2021) was used to estimate the sample size required for this study [16,17,18]. Paired *t*-tests were used to analyze the imaging data. Seventeen subjects were required to achieve a statistical power level of 80%, at a significance level of α = 0.05 and effect size = 0.5. Repeated measures ANOVA was applied to the behavioral data. Fourteen subjects were required to achieve a statistical power level of 80% at a significance level of α = 0.05 and effect size = 0.1. Correlations between behavioral and imaging data were analyzed. Twenty-six subjects were required to achieve a statistical power level of 80% at a significance level of α = 0.05 and effect size = 0.5. Therefore, a sample of 30 subjects was selected.

### 2.7. Statistical Analysis

Behavioral data were collated and statistically analyzed using Stata15(Stata Corp., College Station, TX, USA). Continuous variables with a normal distribution were expressed as “mean ± standard deviation,” while continuous variables without a normal distribution were described as “median (Q25~Q75)”. The data of five PVT indicators that did not conform to normality and the chi-square test were tested by Friedman’s non-parametric repeated measures ANOVA test. For imaging data, paired *t*-tests between acute SD and RW status were performed on the CBF and GMV of each subject, using SPM12. Family-wise error (FWE) correction was used for the repeated comparisons in the paired *t*-test. CBF indicators were correlated with PVT indicators, and Pearson correlation analysis was used when both were quantitative information and met normality. Because the mean RT, fastest 10% RT, minor lapses, major lapses, and false starts do not satisfy a normal distribution, Spearman’s rank correlation analysis was used.

## 3. Results

### 3.1. PVT Change Performance

The five behavioral indicators of these 30 subjects (age: 24.7 ± 2.7 years, BMI: 22.6 ± 2.9 kg/m^2^) were significantly affected after 36 h of SD (Table 1 and Figure 2). Compared to the RW state, the mean RT, fastest 10% RT, minor lapses, and false starts were progressively longer or increased over approximately 20 h of acute SD, while major lapses remained unchanged. Subsequently, each indicator showed a clear trend of fluctuating prolongation or increase, with its peak in the 24 h–32 h range, but all declined at 36 h, with the major lapses returning to normal.

### 3.2. CBF Differences before and after Acute SD

Of the 30 subjects, one was excluded from imaging analysis because of impaired imaging data. Therefore, imaging data from 29 subjects aged 24.7 ± 2.7 years were finally analyzed. Compared to the RW state, the acute SD state had a lower CBF in the left orbital part of the middle frontal gyrus (Frontal_Mid_Orb_L), the left orbital part of the superior frontal gyrus (Frontal_Sup_Orb_L), the left of the rolandic operculum (Rolandic_Oper_L), the left triangular part, and the right opercular part of the inferior frontal gyrus (Frontal_Inf_Tri_L, Frontal_Inf_Oper_R). On the contrary, CBF was increased in the left lingual gyrus (Lingual_L) and the right superior gyrus (Occipital_Sup_R) in the acute SD status, compared to the RW status (paired *t*-test, voxel-level FWE of *p* < 0.05) (Table 2 and Figure 3).

### 3.3. GMV Differences before and after Acute SD

Statistical plots of GMV in subjects (*n* = 29) were imported into Xjview software. There were no significant red or blue stained areas in the statistical actions after 36 h of SD, compared to the RW state, suggesting no significant increase or decrease in GMV (paired *t*-test, *p* < 0.05 for voxel level FWE), as shown in Figure 4.

### 3.4. Correlation between PVT Changes and CBF Changes

A correlation analysis between changes in CBF (*n* = 29) and changes in the five PVT indicators (*n* = 29) was performed, in which only a negative correlation was seen between the lingual_L and the major lapses (*r* = −0.50, *p* < 0.01), a positive correlation was seen between the Frontal_Inf_Oper_R and the false starts (*r* = 0.40, *p* < 0.05), and the Frontal_Sup_Orb_L was positively correlated with the false starts (*r* = 0.41, *p* < 0.05). (Figure 5 and Table 3).

## 4. Discussion

Our study is an objective observation of cognitive behaviors in healthy adults during 36 h of SD, and of brain function and brain structure performance before and after SD. We found regular changes in several indicators of cognitive behavior and several brain areas with reduced or increased CBF. We demonstrated a correlation between cognitive behavioral indicators and CBF changes. Meanwhile, we concluded no significant change in GMV after 36 h of acute SD.

The indicators of mean RT and the number of lapses are susceptible to the decrease in alertness caused by acute SD. They are considered the “gold standard” for assessing the impact of acute SD on cognitive alertness [19]. The five PVT indicators used in this study reveal the effects of acute SD on behavior in various dimensions. The fastest 10% RT reflects changes in vigilance levels [20], and minor lapses and major lapses reflect changes in wakefulness levels [21]. Mean RT and false starts reflect changes in the level of cognitive control. Our study found a flat-rate increase in the four indicators, and no significant change in major lapses for about 20 h after acute SD, but a significant increase in the four indicators after 20 h, and an increase in major lapses, suggesting a dramatic decrease in cognitive performance in adults 20 h after acute SD. Our study found that the four indicators showed a gradual increase within 20 h after SD, and the change of major lapses was not obvious. Still, the four indicators significantly increased after 20 h, and the major lapses also began to rise rapidly. These results indicated that the cognitive ability of adults would decrease sharply after 20 h of SD. A previous study demonstrated that when continuous awakenings exceeded 16 h, most people began to show significant impairments in PVT performance, such as a significant increase in RT, which continued to worsen as awakenings continued [22], consistent with our results for changes in PVT indicators. Furthermore, our trial found that although the overall trend was towards lower levels of alertness, the degree of change during the period fluctuated up and down, possibly due to the influence of circadian rhythms [23]. Still, we believe this circadian influence was not as significant as the effect of acute SD.

ASL-fMRI found that 36 h after SD, the CBF increased in the left lingual gyrus and the right superior gyrus. At the same time, it decreased in the left orbital part of the middle frontal gyrus, the left orbital part of the superior frontal gyrus, the left of the rolandic operculum, the left triangular part, and the right opercular part of the inferior frontal gyrus. The orbital part of the middle frontal gyrus and the orbital part of the superior frontal gyrus are located in the prefrontal cortex and are responsible for performing cognitive functions, and maintaining emotions, thinking, and perception. The triangular and opercular parts of the inferior frontal gyrus are located in Broca’s area, which is the motor speech center and performs motor language functions. The results of reduced CBF in functional brain regions reveal mechanisms of impaired cognitive and motor-verbal function in the brain 36 h after SD, which are also consistent with the results of reduced CBF in the frontal-parietal attention network after acute SD, previously reported by Poudel et al. [24]. It was confirmed by a previous neuroimaging analysis study [25] that the cerebral cortex and, in particular, the frontoparietal attention network, is particularly sensitive to sleep stress. In the present experiment, we also found increased CBF in the left lingual gyrus and the right supraoccipital gyrus, mainly involved in visual information processing [26,27], including face perception and color vision. Previous studies have shown increased visual perfusion [28] after acute SD, along with suppression of mood [29] and hearing [30]. Danyang Kong et al. found that the visual cortex significantly increased CBF after SD, compared to RW [31]. This is consistent with our finding of increased activation of the lingual and supraoccipital gyrus following acute SD, which may represent the need for subjects to remain vigilant to resist the effects of acute SD. In a study that also measured CBF but used positron emission tomography, acute SD ≥ 24 h reduced overall neuronal activity, while thalamic activity was also severely reduced [32]. However, we did not find changes in CBF in the thalamus 36 h after acute SD. It may be that this difference in results is related to differences in experimental method design and conditions, but it is necessary to obtain experimental verification of this by others.

The cognitive reduction caused by acute SD is undoubtedly related to the level of changes in brain function. Tom Cullen et al. found reduced parietal and prefrontal cortex activation during acute SD, corresponding to reduced cognitive function [33]. One study showed that acute SD, combined with reduced amplitude of low-frequency fluctuation in attention-related areas, resulted in reduced attentional function and thus increased perceptual load on visual processing [34]. The present study confirmed the correlation between CBF changes in the visual cortex, prefrontal cortex, and motor speech centers, and PVT indicators.

GMV is an imaging index used to measure the structure of brain tissue [35]. In their study, Sun et al. [13] reported a significant increase in GMV in the right temporal pole 24 h after acute SD. It has also been suggested that acute SD was correlated with extensive GMV changes in cerebral regions, including the thalamus, cerebellum, insula, and parietal cortex [8]. However, our final results found no significant change in GMV in acute SD subjects. Several animal studies have shown that the maintenance of brain cell membranes and myelin sheaths is easily impaired by SD [36]. A previous neuroimaging study suggested that chronic SD may lead to reduced GMV in the ventral medial prefrontal cortex through apoptosis, which may affect grey matter development [6]. Melatonin is protective against CA1 neuronal density in animal models [37], and melatonin production is significantly reduced in chronically SD individuals, resulting in altered structural volumes in structural subregions of the hippocampus [38]. Therefore, we suggest that it should be chronic cumulative SD that causes GMV changes, whereas 36 h acute SD effects do not cause changes in brain structure.

There are several limitations in the present study. First, due to the physiological cycle of female subjects and endocrine disruption, caused by acute SD, only male subjects were recruited, so the results may have a gender bias. Second, no biochemical markers were included. One study claimed that circadian hormone levels and inflammatory markers affected by acute SD could effectively elucidate cellular metabolism, biochemistry, and microenvironment, and effectively assess the extent of brain tissue damage [39]. Biochemical indicators will be added in subsequent studies of the mechanisms underlying imaging and behavioral changes after acute SD. Third, a resting-state BOLD-CBF dynamic coupling study [40], considered an innovative indicator to assess brain health, was not performed. It is also one of the important areas of our future research.

## 5. Conclusions

This study aimed to investigate objective changes in cognitive behaviors, brain function, and brain structure in normal adults following acute SD. The findings showed a decrease in behavioral alertness after 36 h of acute SD in adults, as well as a decrease in CBF in some brain areas that perform cognitive and motor-verbal functions, but an increase in CBF in brain areas that control the visual cortex. In general, we believe that changes in behavioral alertness and CBF are caused after 36 h of acute SD and that changes in behavioral alertness are associated with altered brain function, whereas short-term acute SD does not cause changes in brain structure.

## Figures and Tables

**Figure 1 brainsci-12-01534-f001:**
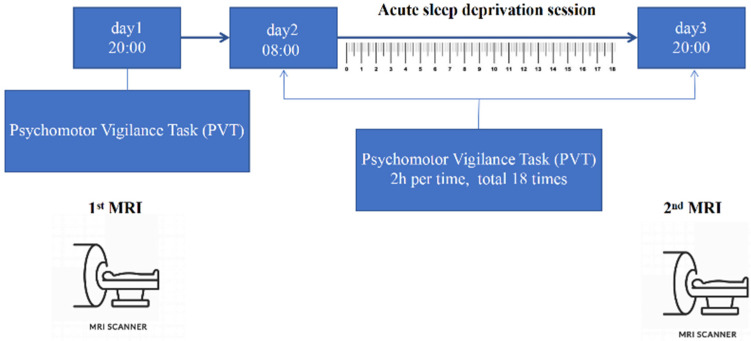
Experimental paradigm of the study.

**Figure 2 brainsci-12-01534-f002:**
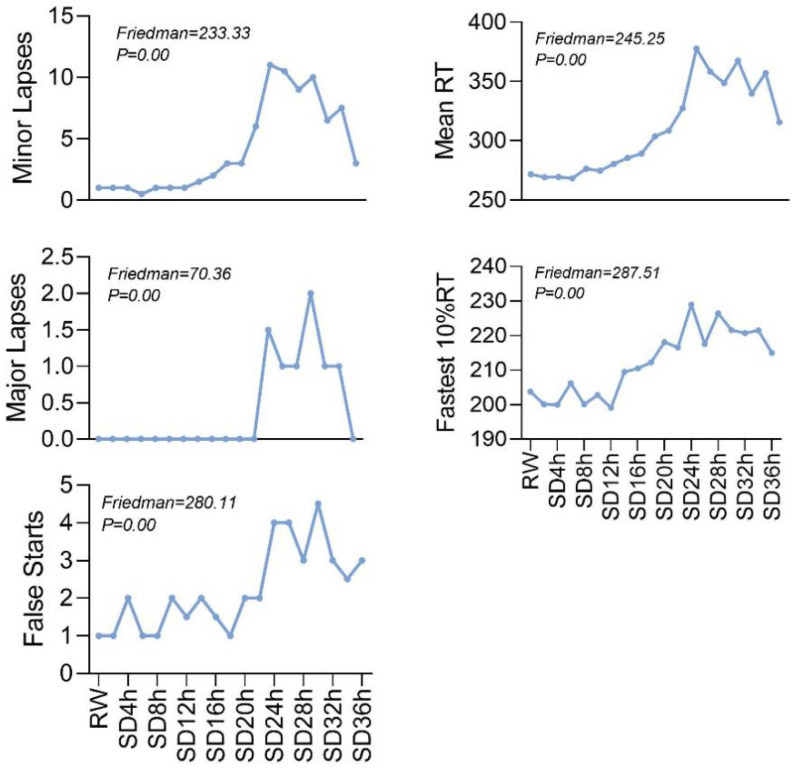
PVT performance at 19 points during the acute sleep deprivation period.

**Figure 3 brainsci-12-01534-f003:**
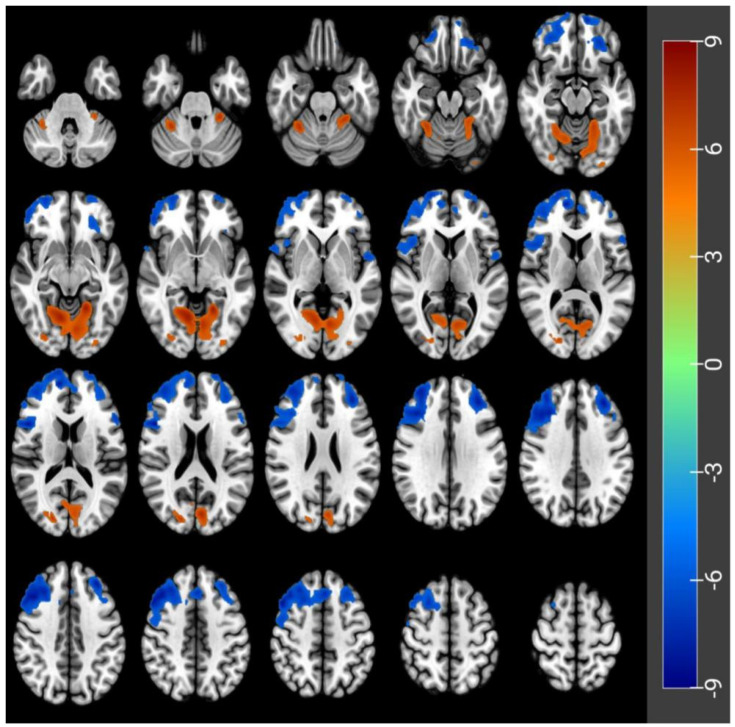
CBF differences before and after acute SD (paired *t*-tests, voxel-level FWE, *p* < 0.05). Hypoperfusion in the Frontal_Mid_Orb_L, Frontal_Inf_Oper_R, Frontal_Sup_Orb_L, Rolandic_Oper_L and Frontal_Inf_Tri_L. Hyperperfusion in the Lingual_L and Occipital_Sup_R.

**Figure 4 brainsci-12-01534-f004:**
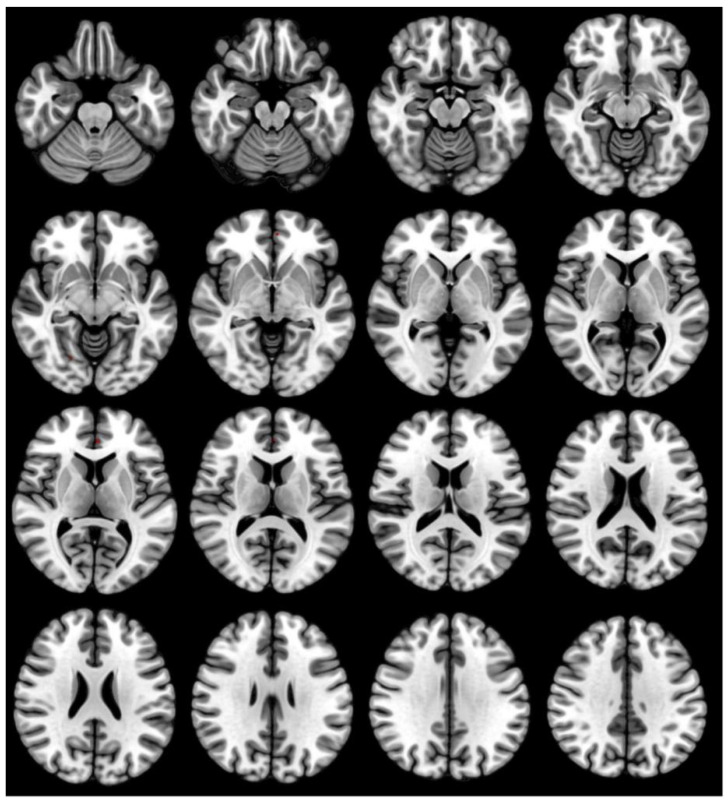
No significant changes in the GMV seen when comparing the RW and acute SD states (paired *t*-tests, voxel-level FWE, *p* < 0.05).

**Figure 5 brainsci-12-01534-f005:**
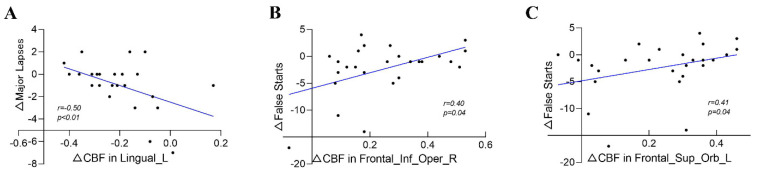
(**A**) Correlation between Lingual_L and major lapses; (**B**) Correlation between Frontal_Inf_Oper_R and false starts; (**C**) Correlation between Frontal_Sup_Orb_L and false starts.

**Table 1 brainsci-12-01534-t001:** PVT performance at 19 points during the acute sleep deprivation period.

Periods	Minor Lapses	Major Lapses	False Starts	Mean RT	Fastest 10% RT
RW	1 (0, 2)	0 (0, 1)	1 (0, 3)	271.77 (243.29, 286.73)	203.75 (187.75, 217.42)
SD2h	1 (0, 2)	0 (0, 0)	1 (0, 3)	269.14 (248.20, 289.85)	200.06 (179.40, 211.73)
SD4h	1 (0, 2)	0 (0, 0)	2 (0, 3)	269.43 (257.78, 285.34)	199.94 (183.78, 211.90)
SD6h	0.5 (0, 2)	0 (0, 0)	1 (0, 3)	268.27 (255.67, 298.42)	206.21 (192.60, 214.10)
SD8h	1 (0, 2)	0 (0, 0)	1 (0, 4)	276.34 (257.18, 296.77)	200.09 (192.90, 217.70)
SD10h	1 (0, 3)	0 (0, 0)	2 (0, 6)	274.60 (264.77, 298.31)	202.77 (183.67, 209.67)
SD12h	1 (0, 2)	0 (0, 0)	1.5 (0, 4)	280.30 (262.20, 296.73)	199.11 (191.30, 218.00)
SD14h	1.5 (1, 3)	0 (0, 0)	2 (1, 3)	285.50 (261.49, 312.90)	209.47 (200.33, 222.60)
SD16h	2 (1, 4)	0 (0, 0)	1.5 (0, 3)	289.02 (269.29, 320.59)	210.45 (195.78, 227.70)
SD18h	3 (1, 7)	0 (0, 1)	1 (0, 3)	303.61 (277.70, 346.80)	212.20 (202.33, 231.33)
SD20h	3 (1, 6)	0 (0, 1)	2 (0, 4)	308.67 (288.99, 328.13)	218.06 (206.70, 224.36)
SD22h	6 (2, 8)	0 (0, 1)	2 (0, 4)	327.42 (301.67, 359.56)	216.55 (204.90, 236.90)
SD24h	11 (5, 16)	1.5 (0, 3)	4 (1, 6)	377.71 (320.55, 438.73)	228.96 (218.44, 237.50)
SD26h	10.5 (4, 18)	1 (0, 5)	4 (1, 7)	358.15 (326.30, 468.54)	217.54 (210.67, 229.13)
SD28h	9 (2, 13)	1 (0, 4)	3 (1, 7)	348.57 (298.82, 413.51)	226.41 (205.90, 241.40)
SD30h	10 (5, 15)	2 (0, 5)	4.5 (2, 8)	367.55 (331.16, 420.60)	221.50 (203.50, 232.78)
SD32h	6.5 (4, 16)	1 (0, 4)	3 (1, 5)	339.68 (310.89, 423.39)	220.72 (205.44, 231.40)
SD34h	7.5 (5, 12)	1 (0, 3)	2.5 (1, 6)	357.11 (312.21, 397.39)	221.43 (207.00, 233.33)
SD36h	3 (1, 10)	0 (0, 2)	3 (1, 5)	315.53 (286.93, 359.24)	214.93 (201.20, 225.20)
Friedman	233.33	70.36	280.11	245.25	287.51
P	0.00	0.00	0.00	0.00	0.00

**Table 2 brainsci-12-01534-t002:** The change of CBF after 36 h of acute SD.

Peak Location	Side	Cluster Size	MNI Coordinates			Peak *t*-Score	*p*-Value Correctedby FWE
			X	Y	Z		
Frontal_Mid_Orb	L	372	−26	40	−18	−7.2	<0.05
Frontal_Sup_Orb	L	2006	−12	62	−18	−9	<0.05
Frontal_Inf_Oper	R	7735	48	14	18	−8.56	<0.05
Frontal_Inf_Tri	L	121	−56	22	14	−7.19	<0.05
Rolandic_Oper	L	116	−54	2	2	−7.07	<0.05
Lingual	L	3910	−20	−52	−8	8.29	<0.05
Occipital_Sup	R	346	20	−84	18	7.06	<0.05

**Table 3 brainsci-12-01534-t003:** Correlation between PVT changes and CBF changes.

	False Starts		Minor Lapses		Major Lapses		Mean RT		Fastest 10% RT	
	*r*	*p*	*R*	*p*	*R*	*p*	*r*	*p*	*r*	*p*
Frontal_Mid_Orb_L	0.25	0.22	0.26	0.19	0.30	0.14	0.29	0.15	0.10	0.63
Frontal_Inf_Oper_R	0.40	0.04	0.23	0.25	0.35	0.08	0.31	0.12	0.03	0.89
Frontal_Sup_Orb_L	0.41	0.04	0.32	0.11	0.21	0.31	0.34	0.09	0.16	0.45
Frontal_Inf_Tri_L	0.35	0.08	0.32	0.11	0.35	0.08	0.39	0.05	0.16	0.44
Rolandic_Oper_L	0.35	0.08	0.32	0.11	0.35	0.08	0.39	0.05	0.16	0.44
Lingual_L	−0.38	0.05	−0.26	0.20	−0.50	0.01	−0.39	0.05	−0.09	0.67
Occipital_Sup_R	−0.39	0.05	−0.26	0.19	−0.27	0.19	−0.16	0.43	0.05	0.82

## Data Availability

The data presented in this study are available on request from the corresponding author.

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
