# Peer review of "Vigilant Attention, Cerebral Blood Flow and Grey Matter Volume Change after 36 h of Acute Sleep Deprivation in Healthy Male Adults: A Pilot Study"

_brainsci, 2022, doi:10.3390/brainsci12111534_

Round 1
Reviewer 1 Report (Previous Reviewer 2)
all issues addressed
Author Response
Thanks
Reviewer 2 Report (Previous Reviewer 1)
we read with interest the pilot study by Zeng et al titled "Cerebral blood flow and grey matter volume change after 36 hours of acute sleep deprivation in healthy male adults: A pilot study" where the authors utilized 3T MRI to study the effect of 36 hrs of acute SD on different brain functional parameters.
the study design and the study in general fit as a pilot study, however, the study outcomes should be written in a more critical mode where there are a lot of limitations. these should be highlighted as the work results may be impacted by the study design and the authors should be discussing them.
Experimental design Comments:
the study included 30 male cohort with no justification for not including a female cohort.
In addition, the work would need a brief power analysis to indicate how the authors reached the number of 30 participants.
Discussion:
the authors have selected to conduct an acute time point of 36 hrs which was shown to affect the CBF but not other parameters such as brain structure, (which makes sense); I would advise that the authors have this acute paradigm and discuss it in contrast to chronic SD and assess how SD, in general, would impact brain structure and by what mechanism as it may be of interest to the readers.
Author Response
Please see the attachment.

Reviewer 3 Report (New Reviewer)
The authors examined the impact of acute sleep deprivation on cognitive performance, as well as cerebral blood flow and grey matter volume in healthy male adults. The topic is very interesting and would be of interest to broad readership of Brain Sciences. However, there are a lot of conceptual, methodological, and statistical issues with the paper. Hence, the paper needs to be substantially improved before being considered for publication.
Abstract: “CBF was hypoperfused in the left orbital part of the middle frontal gyrus, the left orbital part of the superior frontal gyrus, the left of the rolandic operculum, the left triangular part, and the right opercular part of the inferior frontal gyrus. Nevertheless, the left lingual gyrus and the right superior gyrus were hyperperfused” – it is not clear whether these results reflect SD or RW
Introduction
· For instance, the authors reported “Currently, there are some evidence on changes in CBF after acute SD [12,15], including decreased perfusion in the left and right para hippocampal gyrus/fusiform cortex and the right prefrontal cortex [17].” without reporting the implications of decreased perfusion in these areas. Similarly, they reported “One study revealed that acute SD could lead to cerebral atrophy and that grey matter volume (GMV) decreased in the somatosensory areas as the duration of acute SD increased [18]. Another study showed that the grey matter density in the right frontal pole, right superior frontal gyrus, and right middle frontal gyrus significantly increased following 24 hours of acute SD [19].”, again, without deepening their arguments. Therefore, I recommend authors to use more evidence to back their claims in the introduction of the manuscript, which I believe is currently lacking. Thus, I recommend the authors to attempt to deepen the subject of their manuscript, as the bibliography is too concise.
· “In this study, the behavioral changes of the subjects after 36 hours of acute SD were observed” – I think this should be “ will be”
2.2. Experiment paradigm: where did this experiment take place ?
2.5. Cognitive measurement: the PVT task needs much more information. How many trials where there, what type of stimuli were used ? Were there any practice trials ? What was the timing for each screen…etc.
2.6. Sample size: which papers have you used to do these calculations or are these generic ?
3.1. Demographic features and PVT performance:
- If any other demographic variables were recorded such as income, education, BMI….etc. please report.
- I do not think one-way ANOVA is suitable for PVT performance. Repeated measures of ANOVA is better suited to find out the main effects of condition and time as well as the interactions between condition and time.
Discussion & Conclusion: These sections would benefit from some thoughtful as well as in-depth considerations by the authors, because as it stands, it lists down all the main findings of the research, without really stressing the theoretical significance of the study. Authors should make an effort, trying to explain the theoretical implication as well as the translational application of their research.
Round 2
Reviewer 2 Report (Previous Reviewer 1)
Accepted
Author Response
Thanks.
Reviewer 3 Report (New Reviewer)
Thanks for amending the paper accordingly. There are a few points that are still missing.
The PVT task STILL needs much more information. How many trials where there, what type of stimuli were used ? Were there any practice trials ? What was the timing for each screen…etc.
References for sample size estimation using Gpower are not added.
Author Response
Please see the attachment.

This manuscript is a resubmission of an earlier submission. The following is a list of the peer review reports and author responses from that submission.
Round 1
Reviewer 1 Report
we read with interest the article entitled "Exploring changes in cerebral blood flow and gray matter volume after 36h of sleep deprivation in healthy adults" by Zeng et al where the authors aimed to study the regional cerebral perfusion changes and assess whether the gray matter volume (GMV) varied in persons after acute sleep deprivation.
the study has major limitation where only one sex is selected and this may highly deviate the results.
the work is preliminary in nature and may necessitate the use of biochemical markers that would be used to correlate with the MRI finings
I would suggest that this work be changed in title as a Pilot study:
Exploring changes in cerebral blood flow and gray matter volume after 36h of sleep deprivation in healthy Male adults: A Pilot Study
the discussion is so brief and many studies are recent and discuss sleep deprivation and biomarkers should be discussed such as the ones by Stefania Mondello et al .
Minor comments:
the manuscript requires major English corrections, it has verb inconsistencies, run on sentences and punctiotion errors
example: may cause may injure
Reviewer 2 Report
Exploring changes in cerebral blood flow and grey matter volume after 36 hours of sleep deprivation in healthy adults
1. Overview:
a. The paper relates an experiment on measuring brain changes following sleep deprevation.
2. General/Larger Issues:
a. It is difficult to understand some of the central concepts of the paper due primarily to issues around grammatical and other language issues. Recommend English language editor.
3. Minor Issues:
a. Title: I would suggest shortening the title to: Cerebral blood flow and grey matter volume change after 36 hours of sleep deprivation in healthy adults
b. Abstract: rewrite, many are not English language sentences.
c. Methods: Justify sample size
d. Other Minor Issues:
i. The manuscript would also be more readable if the unnecessary acronyms were removed and their usage made consistent
Round 2
Reviewer 1 Report
thank you
Author Response
Thanks for your reply.
Reviewer 2 Report
Point 1: It is difficult to understand some of the central concepts of the paper due primarily to issues around grammatical and other language issues. Recommend English language editor.
Response 1: Thank you for your comments. We have corrected the English language to make it easier for you to understand the content of our articles.
- problems remain
Point 2: I would suggest shortening the title to: Cerebral blood flow and grey matter volume change after 36 hours of sleep deprivation in healthy adults.
Response 2: Thanks for your suggestion. We combined the comments of two reviewers on the title and revised the title to: Cerebral blood flow and grey matter volume change after 36 hours of sleep deprivation in healthy male adults: A Pilot Study.
ok
Point 3: Abstract: rewrite, many are not English language sentences.
Response 3: Thanks for your suggestion. We have rewritten our abstract.
- problems remain
Point 4: Methods: Justify sample size.
Response 4: Thanks for your suggestion. We add a justification for the sample size in the Discussion section. Experiments on sleep studies generally assume that when the sample size is less than 15, the variance and standard deviation of behavioral indicators will be too large, and the 30 sample size we use is a more reasonable and economical design. In addition, we have provided a more precise description of the data collection of the subjects in the revised manuscript.
Original: In the Results section: Among the 30 recruited participants, one was excluded due to impaired imaging data. Therefore, 29 male participants aged 24.7 ± 2.7 years were included in the final analysis.
Revised:(1) In the Results section: Of the 30 enrolled candidates, one was removed from the imaging analysis because of impaired imaging data. Therefore, 29 male subjects of age 24.7 ± 2.7 years were eventually analyzed for imaging data, and 30 male subjects were all involved in the analysis of the behavioral data
(2)In the Discussion section: It is generally believed that the variance and standard deviation of behavioral indicators can be too large when the sample size is less than 15, and the sample size of 30 that we have used is a more reasonable and economical design.
this is not what is meant by justification of sample size, you should do a power analysis using something like GPower to show that you have the number of observations necessary to detect the expected sample size - which was obviously not done - this is an extremely simple and common requirement for studies such as yours otherwise how do you know if your study can do what it says it can do?
Point 5: The manuscript would also be more readable if the unnecessary acronyms were removed and their usage made consistent.
Response 5: Thanks for your suggestion. We have verified the abbreviations and removed unnecessary ones ,such as: electroencephalography(EEG) , the executive control network(ECN), the dorsal attention network(DAN), the salience network(SAN), right parahippocampal gyrus/fusiform cortex (pHipp/Fus), the right prefrontal cortex (PFC), the grey matter density (GMD), the right frontal pole (FP), right superior frontal gyrus (SFG), one-way analysis of variance (ANOVA), the amplitude of low-frequency fluctuations (ALFF), the right temporal pole (TP).
there are many remaining that could be removed to improve readability and you have removed some that are standard and should be left in like EEG and ANOVA
Round 3
Reviewer 2 Report
all issues addrerssed, my only suggestion would be to include a glossary of abbreviations to help readers with the many acronyms still included